# Ethnicity, Age, and Gender Differences in Glycated Hemoglobin (HbA1c) Levels among Adults in Northern and Eastern Sudan: A Community-Based Cross-Sectional Study

**DOI:** 10.3390/life13102017

**Published:** 2023-10-05

**Authors:** Sumia F. Ahmed, Ahmed A. Hassan, Majdolin M. Eltayeb, Saeed M. Omar, Ishag Adam

**Affiliations:** 1Biochemistry Department, Faculty of Medicine, Najran University, Najran 664621, Saudi Arabia; sumia_1978@yahoo.com (S.F.A.); majdo.m.eltayeb@gmail.com (M.M.E.); 2Faculty of Medicine, University of Khartoum, Khartoum P.O. Box 102, Sudan; aa801181@gmail.com; 3Faculty of Medicine, Gadarif University, Gadarif 32211, Sudan; drsaeedomar@yahoo.com; 4Department of Obstetrics and Gynecology, Unaizah College of Medicine and Medical Sciences, Qassim University, Unaizah 56219, Saudi Arabia

**Keywords:** diabetes mellitus, glycated hemoglobin, ethnicity, age, body mass index

## Abstract

Background: The level of association between glycated hemoglobin (HbA1c) level and ethnicity, age, and gender is not yet settled. This study aimed to investigate the association between ethnicity, age, and gender and HbA1c level among adults who were known not to have diabetes mellitus in northern and eastern Sudan. Methods: A comparative community-based cross-sectional study was conducted. Sociodemographic and clinical characteristics data were collected. HbA1c levels were measured, and multiple linear regression analysis was performed. Results: A total of 898 adults (363 in northern Sudan and 535 in eastern Sudan) were included; 349 (38.9%) were males. The HbA1c level was significantly higher in eastern Sudan, and there was no significant difference in HbA1c levels between genders. In multiple linear regression, for adults with HbA1c <6.5%, ethnicity and BMI were associated with HbA1c, but age and gender were not associated with HbA1c. In northern Sudan, age was positively associated with HbA1c, and there was no association between gender, BMI, and HbA1c in adults with HbA1c <6.5%. In eastern Sudan, BMI was positively associated with HbA1c, and there was no significant association between age and gender and HbA1c level in adults with HbA1c <6.5%. Conclusion: HbA1c levels are influenced by ethnicity and age but not by gender.

## 1. Introduction

Glycated hemoglobin (HbA1c) is an important tool to diagnose diabetes mellitus (DM) and to monitor glycemic control and risk for developing complications, especially in settings with limited resources [1,2]. Nevertheless, several factors such as acute and chronic blood loss; hemolytic anemia; splenomegaly; and iron, folate, and vitamin B-12 deficiencies are reported to influence HbA1c levels [3]. Recently, more attention has been paid to the role of ethnicity, age, and gender differences on HbAIc levels [4,5,6,7]. For instance, the influence of gender difference is reported in several adult health conditions such as DM and high body mass index (BMI) [5]. Such influence of gender difference on HbAIc levels is closely linked to age; hence, age and gender should be considered in the application of HbA1c in the diagnosis of DM [4,5,8]. Some studies showed that HbA1c levels for males were significantly higher than for females among all participants [4,8,9].

It has been reported that HbA1c levels are associated with age [4,9]. Variations in HbA1c among age groups have been reported [4,9,10]. In addition to age and gender, HbA1c is also influenced by ethnic differences in people without known DM [6,7,11,12]. A study in Mauritius showed that people of African ethnicity from Rodrigues have higher HbA1c than do those of South Asian ethnicity [11]. In addition, such variation has been noted between Caucasians and Inuits in Denmark as well as between African Americans and Caucasians in the United States [6,7].

Understanding the influence of ethnicity, age, and gender differences on HbAIc is an essential component in any strategy to combat DM and its complications for several reasons. Among them, it will help in standardizing the HbA1c cutoff points for ethnicity, age, and gender to diagnose DM, and having more reliable testing strategies (high sensitivity and specificity) will help in the rational use of resources, especially in resource-limited settings such as Africa. Having specific cutoff points of HbA1c for ethnicity, age, and gender might improve the reliability of HbA1c as a diagnostic and follow-up tool for DM. Although the emerging literature has documented the variations in the HbA1c cutoff levels for the diagnosis of DM for gender and ethnicity [4,11,12], to the best of the authors’ knowledge, no study has investigated the influence of ethnicity, age, and gender differences on HbA1c levels among Sudanese adults, where a high prevalence of DM among adults exists in different regions of Sudan [13,14]. In eastern Sudan, our cross-sectional community-based study of 600 Sudanese adults showed that the prevalences of type 2 DM, newly diagnosed type 2 DM, and uncontrolled type 2 DM were 20.8%, 10.0%, and 80.0%, respectively [13]. Eltom et al. [14], in their cross-sectional population-based study in Northern State and River Nile State in northern Sudan, included 5376 participants from 14 localities and found the following: the prevalence of type 2 DM and prediabetes in northern Sudan has increased significantly over a period of two decades, the proportion of undiagnosed type 2 DM has decreased by 35% compared to the proportion of undiagnosed cases in 1996, and there are ethnic variations in type 2 DM prevalence independent of other risk factors studied, with the highest prevalence occurring among the populations of Egyptian descent and mixed origins compared to the indigenous population of northern Sudan. It is clear from the existing data that DM and its complications represent a major public health problem in Sudan. Use of the HbA1c test is crucial, especially in resource-limited settings such as Sudan. Conducting further research on the potential factors influencing HbA1c such as ethnicity, age, and gender is of paramount importance in order to be more accurate in diagnosing DM, i.e., to explore if there is a difference in HbA1c by ethnicity, age, and gender. Moreover, this study will act as a base for further studies to have enough data on which to update our diagnostic and monitoring strategies for DM, given that Sudan is a large country with multiethnicity. Therefore, the current study aimed to investigate the influence of ethnicity, age, and gender differences on HbA1c levels among adults who were known not to have DM in northern (Almatamah, River Nile State) and eastern (Gadarif) Sudan. These two regions have entirely different ethnicity populations (inhabited by different tribes), and they are at least 600 km from each other.

## 2. Materials and Methods

### 2.1. Study Areas

The details of the study areas have been mentioned in our previous work [13,15]. In summary, a comparative community-based cross-sectional study was conducted during the period from July to September 2022. It was conducted in River Nile State, northern Sudan, and Gadarif State, eastern Sudan, which are two of the 18 states of Sudan. According to the 2008 census, the total populations of River Nile State and Gadarif State were 1,120,441 people and 1,400,000 people, respectively [16]. There are seven localities (the lowest administrative units in Sudan) in the River Nile State. Among them is Almatamah Locality, River Nile State, northern Sudan. Almatamah Locality neighbors Khartoum State and is approximately 100 km from Khartoum, the capital of Sudan. Among the seven localities, Almatamah Locality was chosen as the majority of the population is descended from the Ja’alin tribe. In contrast, the population of Gadarif State includes all Sudanese tribes from all different regions of Sudan.

Gadarif State is located in the east of Sudan, neighboring Ethiopia. The capital of the state is Gadarif City. There are eleven localities in Gadarif State. It has vast land suitable for agriculture, and it is home to the largest projects for rain-fed agriculture in Sudan. It consists of ethnic groups representing different tribes (i.e., a multiethnic society) [17]. Initially, four localities were chosen randomly from the eleven localities. From each of the four localities, the appropriate sample size was recruited according to the locality’s size.

### 2.2. Study Population and Design

From each selected locality, the appropriate sample was selected based on population density to obtain the desired sample size. The first member in each household who agreed to participate and met the study inclusion criteria was selected. The investigators trained ten medical officers in data collection methods in order to standardize the data collection procedure and maintain data quality.

### 2.3. Definitions

Strengthening the reporting of observational studies in epidemiology (STROBE) guidelines were strictly followed [18]. BMI was computed as the weight in kg divided by the square of the height in meters (kg/m^2^). BMI was grouped according to the World Health Organization (WHO) classification: underweight (<18.5 kg/m^2^), normal weight (18.5–24.9 kg/m^2^), overweight (25.0–29.9 kg/m^2^), or obese (≥30.0 kg/m^2^) [19]. HbAIc was categorized based on the International Diabetes Federation (IDF) [1] guidelines into normal HbA1c levels at <6.5% and newly diagnosed diabetes with HbAIc levels at ≥6.5%. Participants who were known diabetics were excluded from the study.

### 2.4. Inclusion Criteria

Sudanese residents (men and women) who were ≥18 years of age and living in a household were chosen using the lottery method. The participants were apparently healthy individuals who were known not to have or to have been treated for DM. If the selected house was uninhabited or its inhabitants refused to participate, the next house was selected to meet the target number for the study.

### 2.5. Exclusion Criteria

Participants whose age was <18 years, adults with previously known diabetes, adults who smoked, pregnant women, patients with known hemoglobinopathy, patients with poor cognitive functions, and severely ill patients were excluded from this study.

### 2.6. Data Collection

For the current data collection, the WHO three-level stepwise approach questionnaire was used [20]. The questionnaire was used to gather data on sociodemographic characteristics, including age in years, gender (male and female), and education level (<secondary and ≥secondary).

### 2.7. Anthropometric Measurements

The participant’s weight was measured in kg using the standard procedure (well-calibrated scales adjusted to zero before each measurement). The participants stood with minimal movement and with their hands by their sides. Moreover, shoes and excess clothing were removed. The participants’ height was then measured in centimeters after standing straight with their backs against a wall and feet together.

### 2.8. Sample Collection and Processing

Each participant provided 3–5 mL of blood for testing HbA1c. HbA1c level was measured using an Ichroma machine in accordance with the manufacturer’s instructions (Chuncheon-si, Gang-won-do, 24398, Republic of Korea), ichroma™. HbA1c is a fluorescence immunoassay (FIA) for the quantitative determination of HbA1c in human whole blood, as previously described in our previous work in eastern Sudan [13].

### 2.9. Sample Size Calculation

OpenEpi Menu was used to compute the desired sample size [21]. A total sample size of 898 participants for the study was calculated as a ratio of 1.5:1 between Gadarif State and River Nile State based on the state populations. More participants were selected from Gadarif State (1.5:1), which is a large agricultural area, unlike River Nile State, where people move to the neighboring Khartoum State. We assumed a difference of 0.2 in the mean HbA1c between the participants in Gadarif State and River Nile State (6.0 vs. 5.8). As this was the first study of its kind conducted in Sudan, our assumption was dependent on the mean HbA1c for people of different ethnicities without known diabetes in Mauritius in a previous study [11]. This sample size was calculated to detect a difference of 5% at *α* = 0.05 with a power of 80%.

### 2.10. Statistical Analysis

Data were entered into a computer using IBM Statistical Package for the Social Sciences^®^ (SPSS^®^) for Windows, version 22.0 (SPSS Inc., New York, NY, USA). The proportions were expressed as frequencies (%). The continuous data were expressed as the mean (standard deviation, SD). Chi-square tests and *t*-tests were used to compare the proportions and means between the two groups. Multiple linear regression was performed with HbA1c levels as the dependent variables and sociodemographic data (age, gender, educational level, and BMI) as the independent variables. A two-sided *p* value of <0.05 was considered statistically significant.

## 3. Results

A total of 898 adults (363 in northern Sudan and 535 in eastern Sudan) were included; 349 (38.9%) and 549 (61.1%) were male and female, respectively. The means (SD) of age, BMI, and HbA1c for the total participants were 44.0 (16.0) years, 27.00 (6.63) kg/m^2^, and 6.41% (1.51%), respectively. There were no significant differences in age or BMI between adults in northern and eastern Sudan. However, there were significantly higher numbers of females and lower levels of education in eastern Sudan. The mean (SD) HbA1c levels were significantly higher in eastern Sudan [6.70 (1.71) % vs. 5.99 (1.01)%, *p* < 0.001], even when HbA1c was compared between females in northern [6.01 (0.99)%] and eastern Sudan [6.61 (1.66)%, *p* < 0.001] and between males in northern [5.96 (1.02)%] and eastern Sudan [6.90 (1.80)%, *p* < 0.001]. Moreover, significantly higher numbers of adults in eastern Sudan had HbA1c levels >6.5% (Table 1; Figure 1).

There was no significant difference in the HbA1c levels between males and females in the total sample, even when categorized HbA1c (<6.5% vs. ≥6.5%) was compared between males and females in each region separately (Table 2 and Table 3). 

In multiple linear regression, while ethnicity (coefficient = 0.22, *p* value < 0.001), age (coefficient = 0.006, *p* value < 0.001), and BMI (coefficient = 0.007, *p* value = 0.002) were positively associated with HbA1c among all studied participants, there was no association between gender and HbA1c level (coefficient = −0.05, *p* value = 0.162). When the total adults were stratified as adults with HbA1c <6.5% or HbA1c ≥6.5%, age and gender were not associated with HbA1c; however, ethnicity was associated with HbA1c in adults with HbA1c <6.5% (coefficient = 0.90, *p* value = 0.034) and in adults with HbA1c ≥6.5% (coefficient = 0.79, *p* value = 0.001). BMI was associated with HbA1c in adults with HbA1c <6.5% (coefficient = 0.007, *p* value *=* 0.037), and it was not associated with HbA1c in adults with HbA1c ≥6.5% (coefficient = −0.014, *p* value = 0.354), Table 2.

In all adults in northern Sudan, age was positively associated with HbA1c level (coefficient = 0.014, *p* value ≤0.001); there was no association between gender (coefficient = 0.16, *p* value = 0.199), BMI (coefficient = −0.004, *p* value = 0.669), and HbA1c level. When participants were stratified as adults with HbA1c <6.5% or HbA1c ≥6.5%, age (coefficient = 0.004, *p* value = 0.016) was positively associated with HbA1c in adults with HbA1c <6.5%, and there was no association between gender, BMI, and HbA1c in adults with HbA1c <6.5%. 

In all adults in eastern Sudan, age (coefficient = 0.023, *p* value < 0.001) and BMI (coefficient = 0.03, *p* value = 0.010) were positively associated with HbA1c level, and there was no significant association between gender and HbA1c level (coefficient = −0.26, *p* value *=* 0.104). When participants were stratified as adults with HbA1c <6.5% or HbA1c ≥6.5%, BMI was positively associated with HbA1c <6.5% (coefficient = 0.011, *p* value = 0.012); there was no association between age and gender in adults with HbA1c <6.5%. Age was positively associated with HbA1c ≥6.5% (coefficient = 0.021, *p* value = 0.025); there was no association between gender (coefficient = −0.02, *p* value = 0.932) or BMI (coefficient = −0.011, *p* value = 0.530) and HbA1c ≥6.5%, Table 3.

## 4. Discussion

The current study showed that adults without a prior diagnosis of DM in eastern Sudan have higher HbA1c levels compared to their counterparts in northern Sudan and that HbA1c levels are associated with ethnicity and age but not with gender among adults in Sudan. The influence of ethnicity on DM has previously been reported in northern Sudan; a cross-sectional population-based study in River Nile State and Northern State included 5376 participants and showed that, in comparison with the indigenous population (Nubians), individuals of Egyptian descent (those whose roots belong to Egypt) and those of mixed origin (other Sudanese tribes) had an increased risk of type 2 DM [14]. Likewise, the effect of ethnicity on HbA1c levels in individuals without DM has been reported in several studies including in Africa [6,7,11,12,22]. A population-based noncommunicable disease survey included 6701 adults without known DM (from different ethnicities) from Mauritius and revealed that people of African ethnicity from Rodrigues have higher HbA1c levels than do those of South Asian ethnicity [11]. The National Diabetes Survey of Pakistan (NDSP) 2016–2017, which included 6836 individuals without prior known diabetes, recommended considering modification in cutoff points by ethnicity or region based on relevant community-based epidemiological surveys in order to validate HbA1c as a diagnostic tool [23]. A recent meta-analysis that included twelve studies involving 49,238 individuals showed that, in individuals without DM, HbA1c levels are higher in Blacks, Asians, and Latinos when compared to White persons [22]. 

The influence of ethnicity on HbA1c levels could be attributed to several reasons, including variations in the rate of erythrocyte turnover and the rate of protein glycation among ethnicities [11]. Hemoglobinopathy such as thalassemia syndromes can alter HbA1c as it can contribute to the glycation gap in Blacks [24]. In addition, variation in insulin resistance was reported among ethnicities [25]. Race and ethnicity could influence HbA1c via the inflammatory processes that mediate via diet quality, BMI, and C-reactive protein (CRP) [26]. The high HbA1c levels in eastern Sudan compared to those in northern Sudan could be attributed not only to ethnicity but also to other sociodemographic factors. These ethnic disparities in HbA1c levels could also be explained by the low education level in eastern Sudan (i.e., 66.9% in northern Sudan and 59.1% in eastern Sudan were educated). The disparities could be attributed to differences in the access to health care between the two studied populations (i.e., less access to health care in eastern Sudan and, as a consequence, high HbA1c). Our experiences in eastern Sudan showed late presentation of diabetic patients to health care, sometimes even with diabetic complications such as diabetic foot.

Moreover, previous studies reported a high prevalence of iron deficiency anemia in eastern Sudan [27], which might elevate HbA1c [3]. Such differences in HbA1c among ethnicities might impact using the current HbA1c cutoff points to diagnose DM in all ethnic populations. Moreover, these differences address the importance of mapping HbA1c by ethnicity in Sudan, aiming for more precision in diagnosing DM using HbA1c as a diagnostic and follow-up tool. Therefore, sociodemographic factors, lifestyle habits, and anthropometric characteristics must be considered in the analysis of studies about HbA1c cutoff levels because they may influence HbA1c levels.

The present study showed age was positively associated with HbA1c levels in adults both across regions and within northern and eastern Sudan. Several studies showed similar results in participants with no prior diagnosis of DM and called for a consideration of age when selecting HbA1c as a criterion for diagnosing DM [8,10,28]. The positive association between age and HbA1c is not only reported in adult populations but also has been reported among healthy children and adolescents [29]. For instance, a population-based cohort study that included 2,455 healthy German children and adolescents aged between 0.5 and 18 years showed that age, male gender, socioeconomic status, pubertal stage, and BMI were associated with higher HbA1c levels [29]. The positive association between age and HbA1c could be attributed to several reasons, including the strong association between age and BMI [30]. In general, the present study showed an increase in HbA1c levels with age in all studied populations and in each studied region, reaching its peak in the age category of 50 to 59 years. Huang et al. observed no association between HbA1c levels for ages between 50 and 70 years in males, but this did not occur in females [4]. 

The present study showed a positive association between BMI and HbA1c in all studied participants and within eastern Sudan. In line with this result, previous studies showed a similar result [9,30]. For example, a retrospective cohort study of 211,833 healthy adult Chinese revealed a linear association between BMI and the risk of incident DM [30]. This indicates that HbA1c level is affected by age–BMI interaction, and that age and BMI should be considered in the application of HbA1c in the diagnosis of DM.

Although the current study showed significant differences between age and ethnicity and HbA1c, there was no significant difference between HbA1c levels and gender. This result is similar to previous studies that revealed that HbA1c levels are not influenced by gender [28]. On the contrary, other studies found the opposite, i.e., associations between HbA1c levels and gender in adults without a prior diagnosis of DM in China [8] and Taiwan [4]. For example, some studies showed a strong association between HbA1c levels and males [4,8,9]. The Taiwan Biobank database analysis of a total of 4748 adult Taiwanese participants (2183 males and 2563 females) concluded that age and gender are important factors affecting HbA1c levels [4].

The contradictory data regarding the influence of gender on HbA1c levels could be explained by gender lifestyle rather than by sex per se [31,32,33]. In support of our explanation, while women with DM had worse glycemic control than men in Brazil and Venezuela [34], it was the opposite in Nigeria [32]. A longitudinal population-based study in China of 334 participants with prediabetes concluded that physical activity in males and waist circumference in females are important factors predicting both progression to diabetes and regression to normal glucose regulation [31]. Moreover, Harreiter and Kautzky-Willer addressed the role of sex and gender differences in the prevention of DM; they called for a more gender-sensitive clinical approach, with consideration of ethnicity to improve quality of life, health, and life expectancy in both sexes with a high risk for subsequent progression to DM [33]. The lack of association between HbA1c and gender in the current study might reflect the complexity of gender’s interaction with other factors, including psychosocial [35] and sociocultural factors [33].

The current study provided valuable information regarding the influencing factors of ethnicity, age, and gender on HbA1c levels, and its findings will be used to design further studies; however, there are some limitations to this study that need to be mentioned. The nature of the present study was cross-sectional. A longitudinal study will provide more clarity regarding such an association between these factors and HbA1c. Although our study was community-based in two regions of Sudan, the present study does not provide nationally representative data on the distribution of HbA1c levels in the Sudanese population because it focused on two geographical regions (northern and eastern Sudan).

Although this survey was community-based, an element of bias may still be present. Additionally, the study lacks data for younger ages (<18 years). Our study did not collect other information about fasting plasma glucose, the oral glucose tolerance test, or hemoglobin; collecting such information is crucial to future studies in order to know the reliability (the exact cutoff points) of HbA1c in diagnosing DM. In our follow-up studies, all these limitations will be addressed in order to explore the temporal relationships between age, ethnicity, gender, and HbA1c levels in Sudanese adults, aiming to bring more detail to the present findings to help us as health care professionals to adapt/implement our current diagnostic and monitoring strategies for diabetes.

## 5. Conclusions

Our results showed that HbA1c levels are influenced by ethnicity and age but not by gender. Further studies are recommended to identify age- and ethnicity-adjusted HbA1c cutoff levels in clinical practice in order to improve the diagnosis and prognosis of DM among the Sudanese population.

## Figures and Tables

**Figure 1 life-13-02017-f001:**
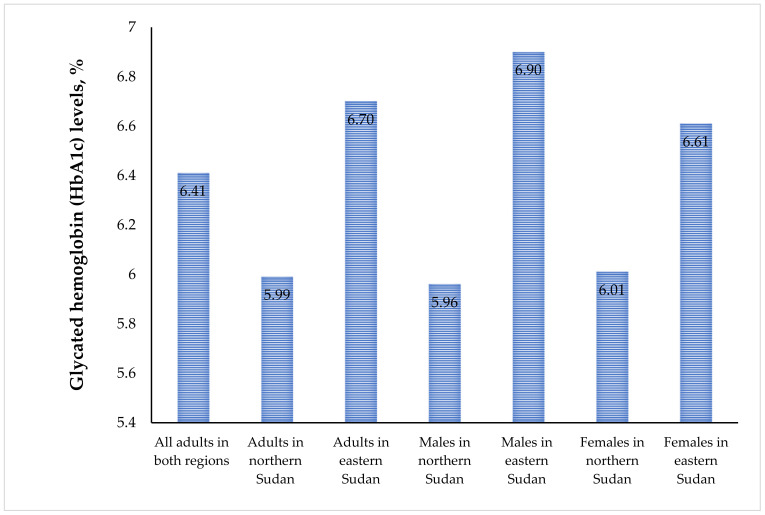
Glycated hemoglobin (HbA1c) levels among adults (males and females) in northern and eastern Sudan.

**Table 1 life-13-02017-t001:** Characteristics of the adults (males and females) in northern and eastern Sudan.

Variable		Total (898)	Northern Sudan (363)	Eastern Sudan (535)	*p* Value
		Mean (Standard Deviation)			
Age, years		44.0 (16.0)	44.7 (15.7)	43.54 (16.2)	0.292
Body mass index, kg/m^2^		27.00 (6.63)	26.61 (6.30)	27.27 (6.83)	0.140
Glycated hemoglobin (HbA1c) for both sexes		6.41 (1.51)	5.99 (1.01)	6.70 (1.71)	<0.001
HbA1c for all females		6.42 (1.51)	6.01 (0.99)	6.61 (1.66)	<0.001
HbA1c for all males		6.39 (1.51)	5.96 (1.02)	6.90 (1.80)	<0.001
		Frequency (Proportion)			
Gender	Male	349 (38.9)	189 (52.1)	160 (29.9)	<0.001
	Female	549 (61.1)	174 (47.9)	375 (70.1)	
Education level	≥ secondary	559 (62.2)	243 (66.9)	316 (59.1)	0.017
	< secondary	339 (37.8)	120 (33.1)	219 (40.9)	
Obesity	No	625 (69.6)	259 (71.3)	366 (68.4)	0.347
	Yes	273 (30.4)	104 (28.7)	169 (31.6)	
HbA1c	<6.5%	615 (68.5)	295 (81.3)	320 (59.8)	<0.001
	≥6.5%	283 (31.5)	68 (18.7)	215 (40.2)	

**Table 2 life-13-02017-t002:** Multiple linear regression analysis of factors associated with glycated hemoglobin (HbA1c) among adults (males and females) in the total participants in northern and eastern Sudan.

Variable		All Studied Participants (*n* = 898) HbA1c		HbA1c <6.5% Participants		HbA1c ≥6.5% Participants	
		Coefficient (Standard Error)	*p* Value	Coefficient (Standard Error)	*p* Value	Coefficient (Standard Error)	*p* Value
Age, years		0.006 (0.001)	<0.001	0.001 (0.001)	0.246	0.012 (0.007)	0.102
Body mass index, kg/m^2^		0.007 (0.002)	0.002	0.007 (0.003)	0.037	−0.014 (0.015)	0.354
Gender	Male	Reference					
	Female	−0.05 (0.03)	0.162	−0.03 (0.05)	0.583	0.03 (0.22)	0.883
Education level	≥secondary	Reference					
	<secondary	−0.03 (0.03)	0.652	0.01 (0.05)	0.946	0.12 (0.22)	0.608
Location	Northern Sudan	Reference					
	Eastern Sudan	0.22 (0.03)	<0.001	0.9 (0.04)	0.034	0.79 (0.25)	0.001

**Table 3 life-13-02017-t003:** Multiple linear regression analysis of factors associated with glycated hemoglobin (HbA1c) among adults (males and females) in northern and eastern Sudan.

Variable		Northern Sudan						Eastern Sudan					
		HbA1c		Hba1c <6.5%		HbA1c ≥6.5%		HbA1c		HbA1c <6.5%		HbA1c ≥6.5%	
		Coefficient (Standard Error)	*p* Value	Coefficient (Standard Error)	*p* Value	Coefficient (Standard Error)	*p* Value	Coefficient (Standard Error)	*p* Value	Coefficient (Standard Error)	*p* Value	Coefficient (Standard Error)	*p* Value
Age, years		0.014 (0.003)	<0.001	0.004 (0.002)	0.016	−0.014 (0.012)	0.249	0.023 (0.005)	<0.001	−0.001 (0.002)	0.524	0.021 (0.009)	0.025
Body mass index, kg/m^2^		−0.004 (0.009)	0.669	0.001 (0.04)	0.873	−0.039 (0.028)	0.177	0.03 (0.01)	0.010	0.011 (0.004)	0.012	−0.011 (0.018)	0.530
Gender	Male	Reference											
	Female	0.16 (0.12)	0.199	0.08 (0.06)	0.204	0.31 (0.38)	0.421	−0.26 (0.16)	0.104	−0.12 (0.07)	0.073	−0.02 (0.26)	0.932
Education level	≥secondary	Reference											
	<secondary	0.15 (0.11)	0.184	0.01 (0.06)	0.850	0.22 (0.37)	0.553	−0.14 (0.16)	0.378	0.05 (0.06)	0.397	−0.03 (0.29)	0.919

## Data Availability

The data will be available upon request from the correspondence author.

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
