# Peer review of "Ethnicity, Age, and Gender Differences in Glycated Hemoglobin (HbA1c) Levels among Adults in Northern and Eastern Sudan: A Community-Based Cross-Sectional Study"

_life, 2023, doi:10.3390/life13102017_

Round 1
Reviewer 1 Report
This paper investigates the influence of ethnicity, age, and gender differences on HbA1c levels among Sudanese adults without known DM in northern and eastern Sudan. The study found that HbA1c levels were significantly higher in eastern Sudan compared to northern Sudan. Age and ethnicity were positively associated with HbA1c levels, but there was no significant association between gender and HbA1c levels. The study suggests that variations in HbA1c levels based on ethnicity and age should be considered when diagnosing DM and highlights the need for specific cutoff points per ethnicity, age, and gender.
1. What specific gaps in the existing literature does this study address regarding the influence of ethnicity, age, and gender on HbA1c levels, especially in the context of Sudan?
2. How representative is the sample of adults from northern and eastern Sudan? Were there any selection biases that might affect the generalizability of the findings?
3. Could you provide more details about the methods used to measure HbA1c and the specific criteria for categorizing participants into different HbA1c levels?
4. How should healthcare practitioners in Sudan and similar settings incorporate the findings of this study into their diagnostic and monitoring strategies for diabetes?
5. Given the cross-sectional nature of this study, are there any plans for follow-up studies to explore the temporal relationships between age, ethnicity, gender, and HbA1c levels in Sudanese adults?
Author Response
We would like to thank the editor and reviewers for their valuable comments. We believe these comments dramatically improved the quality of our manuscript.
Our responses to the raised comments are inserted in the revised manuscript in coloured text. Moreover, we provide an author response letter with a point-by-point annotated response to all raised comments from the editor and reviewers.
Reviewer
This paper investigates the influence of ethnicity, age, and gender differences on HbA1c levels among Sudanese adults without known DM in northern and eastern Sudan. The study found that HbA1c levels were significantly higher in eastern Sudan compared to northern Sudan. Age and ethnicity were positively associated with HbA1c levels, but there was no significant association between gender and HbA1c levels. The study suggests that variations in HbA1c levels based on ethnicity and age should be considered when diagnosing DM and highlights the need for specific cutoff points per ethnicity, age, and gender.
- What specific gaps in the existing literature does this study address regarding the influence of ethnicity, age, and gender on HbA1c levels, especially in the context of Sudan?
Response
To our knowledge this is the first study of it is kind that addressed the influence of ethnicity, age, and gender on HbA1c levels, in the context of Sudan. This study will act as a base for further study to tackle this vital issue giving that Sudan is a large country with multiethnicity. (Moreover, this study will act as a base study for further studies to have enough data on which we can update our diagnostic and monitoring strategies for DM, giving that Sudan is a large country with multiethnicity). Please see line 60-62
- How representative is the sample of adults from northern and eastern Sudan? Were there any selection biases that might affect the generalizability of the findings?
Response
We try as much as possible to minimize the selection biases, therefore, we selected Almatamah Locality, River Nile State hence, the majority of population are from one ethnicity. Moreover, based on the larger population in Gadarif, eastern Sudan, larger sample was selected from eastern Sudan i.e. a ration of 1.5 to 1 from Gadarif and Almatamah, respectively.
(More participants were selected from Gadarif State as we assumed more people moved to settle in Gadarif State from other states for agricultural activities, unlike River Nile State where people move to the neighbouring Khartoum State.) please see lines 134-136.
However, elements of bias can to be excluded totally and this point has been mentioned as one of the limitations of the study. Please see line 292.
- Could you provide more details about the methods used to measure HbA1c and the specific criteria for categorizing participants into different HbA1c levels?
Response
More details are added about the methods used to measure HbA1c and the specific criteria for categorizing participants into different HbA1c levels. Please see the method section.
(HbAIc was categorized based on the International Diabetes Federation (IDF) [1] into normal HbA1c levels <6.5%, and newly diagnosed diabetes with HbAIc levels ≥6.5%.). (ichromaMT (HbA1c is a fluorescence Immunoassay (FIA) for the quantitative determination of HbA1c in human whole blood). Please see lines 101-102
- How should healthcare practitioners in Sudan and similar settings incorporate the findings of this study into their diagnostic and monitoring strategies for diabetes?
Response
As this is first study that addressed this topic and gave us initial results, we are very keen to conduct further study that taking into account all the limitations of the current study, aiming to give more details to the current findings to help us as healthcare professionals to adapt our current diagnostic and monitoring strategies for diabetes. (In our follow-up studies, all these limitations will be taken to explore the temporal relationships between age, ethnicity, gender, and HbA1c levels in Sudanese adults, aiming to give more details that to the present findings to help us as healthcare professionals to adapt our current diagnostic and monitoring strategies for diabetes.)Please see line 297-298
- Given the cross-sectional nature of this study, are there any plans for follow-up studies to explore the temporal relationships between age, ethnicity, gender, and HbA1c levels in Sudanese adults?
Response
Thank you for raising this crucial point. As we mentioned above, currently we are planning to conduct further study that taking into account all the current limitations.
(In our follow-up studies, all these limitations will be taken to explore the temporal relationships between age, ethnicity, gender, and HbA1c levels in Sudanese adults, aiming to give more details that to the present findings to help us as healthcare professionals to adapt our current diagnostic and monitoring strategies for diabetes). Please see lines 297-298
Regards
Reviewer 2 Report
Thank you for the opportunity to review this manuscript. It presents an interesting study about the influence of ethnicity, age and gender differences on HbA1c levels among adults who were not known to have DM in northern and eastern Sudan.
The methodological approach is adequate.
The introduction and discussion present a clarifying revision of the literature. However, I have some comments and suggestions presented below.
General comments
Please, review the whole text for typos. For example, line 34. “HbA1clevels”.
P value in the whole manuscript must be expressed as p value.
Extensive English editing in the discussion section.
Specific Comments
2. Materials and Methods
Line 126. “898 participants for the total study was calculated as a ratio of 1.5:1 between the Gadarif State and the River Nile State based on the state population” The proportion of the population between the two states is not. 1.5, is more 1.25. Why you choose 1.5 for the stratification. Please, clarify.
3. Results
Line 158. “There was no significant difference in the HbA1c levels between males and females, even when the HbA1c was compared between males and females in each region separately”. Please, syntax review. In the regression analysis, you are not comparing exactly HbA1c levels between males and females. You are pondering the influence of gender, together with the other independent variables, in the HbA1c levels, and you obtained that gender had no influence, but you have not made this comparison or studied this relation isolated.
Table 3. Last row. “P= 00.553” Please correct to: p=0.553
Line 178: “there was no association between 178 gender, BMI and HbA1c in adults with HbA1c <6.5%” Please correct to: “there was no association between age, gender or BMI in adults with HbA1c ≥6.5%”
4. Discussion
Line 194. “The 194 cross-sectional, population-based study in River Nile State and Northern State included 195 5,376 participants showed in comparison with indigenous population, individuals of 196 Egyptian descents and mixed origin had increased risk of type 2DM.” Please syntax review.
Line 218. “It could be attributed to differences in the access to health care among the two studied population i.e., less access to health care in eastern Sudan and as consequence high HbA1c.” Please, add some data, from another studies or from your expertise that can support the fact that there is a potential lesser access to health care in eastern Sudan.
Line 226. Please, consider add the next reflection: sociodemographic factors, life style habits and anthropometric characteristics must to be considered in the analysis in studies about HbA1c cutoff levels because they may influence HbA1c levels.
Line 234. Which gender in this study (29) was associated with higher HbA1c levels?
Extensive editing of English language required
Author Response
We would like to thank the editor and reviewers for their valuable comments. We believe these comments dramatically improved the quality of our manuscript.
Our responses to the raised comments are inserted in the revised manuscript in coloured text. Moreover, we provide an author response letter with a point-by-point annotated response to all raised comments from the editor and reviewers.
Thank you for the opportunity to review this manuscript. It presents an interesting study about the influence of ethnicity, age and gender differences on HbA1c levels among adults who were not known to have DM in northern and eastern Sudan.
The methodological approach is adequate.
The introduction and discussion present a clarifying revision of the literature. However, I have some comments and suggestions presented below.
Response
Thank you for time to review our article and for your valuable comments
General comments
Please, review the whole text for typos. For example, line 34. “HbA1clevels”.
Response
The article was revised all typos were corrected.
P value in the whole manuscript must be expressed as p value.
Response
The reviewer advice was followed and necessary corrections were done.
Extensive English editing in the discussion section.
Response
The article was edited by another personal to improve its quality.
Specific Comments
- Materials and Methods
Line 126. “898 participants for the total study was calculated as a ratio of 1.5:1 between the Gadarif State and the River Nile State based on the state population” The proportion of the population between the two states is not. 1.5, is more 1.25. Why you choose 1.5 for the stratification. Please, clarify.
Response
We used 1.5 instead of 1.25, as it was observed in the resent years more displaced from other states to Gadarif State, unlike the River Nile State.
(More participants were selected from Gadarif State as we assumed more people moved to settle in Gadarif State from other states for agricultural activities, unlike River Nile State where people move to the neighboring Khartoum State.) Please see lines 133-134
- Results
Line 158. “There was no significant difference in the HbA1c levels between males and females, even when the HbA1c was compared between males and females in each region separately”. Please, syntax review.
Response
It is reviewed. (There was no significant difference in the HbA1c levels between males and females in the total sample size, even when categorizing HbA1c (<6.5% vs. ≥6.5%) was compared between males and females in each region separately, Table 2, 3). Please see line 167.
In the regression analysis, you are not comparing exactly HbA1c levels between males and females. You are pondering the influence of gender, together with the other independent variables, in the HbA1c levels, and you obtained that gender had no influence, but you have not made this comparison or studied this relation isolated.
Response
We are considering these suggestions as requested. Please see lines 166-167
Table 3. Last row. “P= 00.553” Please correct to: p=0.553
Response
It was corrected to (0.553)
Line 178: “there was no association between 178 gender, BMI and HbA1c in adults with HbA1c <6.5%” Please correct to: “there was no association between age, gender or BMI in adults with HbA1c ≥6.5%”
Response
It was corrected to: “there was no association between gender (coefficient = -0.02, p value = 0.932), or BMI (coefficient = -0.011, p value = 0.530), and HbA1c ≥6.5%).
- Discussion
Line 194. “The 194 cross-sectional, population-based study in River Nile State and Northern State included 195 5,376 participants showed in comparison with indigenous population, individuals of 196 Egyptian descents and mixed origin had increased risk of type 2DM.” Please syntax review.
Response
It was reviewed (the cross-sectional, population-based study in River Nile State and Northern State included 5,376 participants showed in comparison with indigenous population (Nubians), individuals of Egyptian descents (those their roots belong to Egypt) and mixed origin (other Sudanese tribes) had increased risk of type 2DM [14].)
Line 218. “It could be attributed to differences in the access to health care among the two studied population i.e., less access to health care in eastern Sudan and as consequence high HbA1c.” Please, add some data, from another studies or from your expertise that can support the fact that there is a potential lesser access to health care in eastern Sudan.
Response
Further data are added regarding access to health care in eastern Sudan. (Our experiences from eastern Sudan showed late presentation of diabetic patients even sometimes with diabetic complications such as diabetic foot to health care.)
Line 226. Please, consider add the next reflection: sociodemographic factors, life style habits and anthropometric characteristics must to be considered in the analysis in studies about HbA1c cutoff levels because they may influence HbA1c levels.
Response
It was added. (Therefore, sociodemographic factors, life style habits and anthropometric characteristics must to be considered in the analysis in studies about HbA1c cut-off levels because they may influence HbA1c levels.)
Line 234. Which gender in this study (29) was associated with higher HbA1c levels?
Response
The gender is added. (male gender)
Round 2
Reviewer 2 Report
I want to thanks the authors for the implementation of the required corrections.
Some typos and minor syntax corrections are still required.